# Reproducibility Study of "Studying How to Efficiently and Effectively Guide Models with Explanations"

## Abstract

This paper investigates the reproducibility of the work presented by Rao et al. (2023) that explores how to guide models efficient and effectively using bounding boxes. Model guidance, achieved by jointly optimizing models with a classification and localization loss, ensures that models are right for the right reasons. Our findings indicate that the results from the original paper are reproducible. However, we observe that while these guided models attend better to the relevant objects, they do so by trading off classification performance. Furthermore, we find that some classes are attended to better than others.

## 1 Introduction

While deep neural networks have excelled in diverse research domains, there is no assurance that these models are learning the correct features (Ribeiro et al., 2016). Certain models may rely on spurious correlations in their predictions, such as incorrectly attending to the background. This leads to unfair and inexplicable decisions and consequently limits a model's ability to generalize (Ghosal & Li, 2023).

To inspect if models rely on spurious correlations in their decision-making, attribution methods have been developed. When guidance from attribution methods is integrated alongside a classification model, the model tends to be more often "right for the right reasons" by steering it towards relevant features (Ross et al., 2017).

Studies on guided models showcase promising results, yet there exists a research gap concerning the effective implementation of model guidance using bounding boxes. Rao et al. (2023) address this gap by conducting a comprehensive comparison of various model guidance methods. This comparison covers various aspects including model architecture, guidance depth, attribution methods, and utilized loss functions. Additionally, the authors propose the use of the Energy Pointing Game (EPG) metric (Wang et al., 2020) and corresponding Energy loss instead of the Intersection over Union (IoU) score and L1 loss that are more commonly used to evaluate and optimize a model's localization performance.

## 2 Scope of reproducibility

This study aims to replicate the original paper of Rao et al. (2023), and investigates the reproducibility of the main claims in the paper, which can be summarized as follows:

- **Claim 1:** Although being relatively coarse, bounding box supervision can provide sufficient guidance to the models whilst being much cheaper to obtain than semantic segmentation masks.

- **Claim 2:** The proposed EPG score constitutes a good loss function for model guidance, particularly when using bounding boxes.

- **Claim 3:** Model guidance can be accomplished cost-effectively by using annotation masks that are noisy or are available for only a small fraction of the training data.

- **Claim 4:** Model guidance with a small number of annotations can improve model generalization under distribution shifts at test time.

## 3 Model guidance

Uncovering the black-box models used throughout the deep learning domain can improve our understanding of the outstanding results these models achieve and can increase the social acceptance of these models by enhancing their transparency for users. Among the approaches to explain decisions in the vision domain are attribution methods (Simonyan et al., 2013). Attribution methods create heatmaps that highlight regions that are important for the model's decision. These attribution maps reveal whether a model is learning object features or overfits to spurious correlations such as common background features (Xiao et al., 2020).

Two examples of attribution methods are IxG and B-cos. **IxG** (Shrikumar et al., 2017) extracts a feature map at a specified layer. Then gradients between the logits and this feature map are calculated. These gradients are multiplied by the feature map and the channel dimensions are summed. The resulting 2D image is upscaled to the original image resolution using bilinear interpolation creating an attribution map. **B-cos** (Böhle et al., 2022) attributions function similarly but use the B-cos network layers, aligning the attribution map to a dynamic weight matrix during training.

Model guidance (Fel et al., 2022) can be used to regularize model explanations by guiding their attributions, improving performance, and making them "right for the right reasons"(Ross et al., 2017). This form of regularization involves extracting an attribution map at a specified layer of the model, which is used to compute a localization loss. The model is guided to look at relevant features by jointly optimizing the model's loss function $\mathcal{L} = \mathcal{L}_{\text{class}} + \lambda_{\text{loc}}\mathcal{L}_{\text{loc}}$ for both the classification loss ($\mathcal{L}_{\text{class}}$) and localization loss ($\mathcal{L}_{\text{loc}}$). Here the importance of localization loss is determined by $\lambda_{\text{loc}}$.

In order to accurately calculate a localization loss segmentation masks are often utilized (Song et al., 2018). However, the decision to explore the use of bounding boxes for model guidance instead of segmentation masks is due to the fact that segmentation masks are time-consuming and thus costly to obtain.

In the original study, the binary cross entropy loss is employed as the classification loss, and four different localization loss functions are compared. To begin with, **PPCE** (Shen et al., 2021) calculates the per-pixel binary cross entropy loss between a bounding box mask and an attribution map, maximizing attributions inside the mask. **RRR\*** (Ross et al., 2017) instead regulates the model to suppress attribution outside the bounding box mask. The $L_1$ **loss** (Gao et al., 2022) minimizes the difference between the mask and the attribution map, ensuring that the model is uniformly guided towards the bounding box while suppressing pixels outside of the bounding box. Lastly, the authors propose the **Energy loss** which is the negative value of the EPG score (Selvaraju et al., 2017), as described in 4.2. Unlike the L1 loss, the Energy loss does not guide to model to attend uniformly within the bounding box. For more details see Appendix E.

## 4 Methodology

The original paper by Rao et al. (2023) came with an extensive codebase that allowed for an extensive evaluation of different model guidance architectures. This included the implementation of different model backbones, attribution methods which can be applied at different layers, and localization loss functions. However, certain crucial parts were missing from the codebase and have thus been added. These additions include code for the waterbirds experiment, functions for visualizing attribution maps, and a comprehensive evaluation function for both models and Pareto-fronts on the test set. Furthermore, in order to support our additional experiments we have added code to evaluate localization performance by class, calculate multiple EPG metrics based on segmentation masks, and added the option to calculate an adaptive IoU threshold for each image. Code available at Github.

### 4.1 Datasets

Two datasets are utilized in this reproduction study. The primary dataset is the PASCAL VOC 2007 (VOC) dataset (Everingham et al., 2007), which contains 9,963 images across 20 different classes with 24,640 annotated object labels and their corresponding bounding boxes. Furthermore, this dataset was extended with segmentation masks for a subset of the images to be used in the newly developed metrics as described in 4.2.

The second dataset is the Waterbirds dataset, which is constructed by cropping out birds from images in the Caltech-UCSD Birds-200-2011 (CUB) dataset (Wah et al., 2011) and transferring them onto backgrounds from the Places dataset (Zhou et al., 2016). Each image in the Waterbirds dataset is labeled as one of two classes: 'waterbird' or 'landbird'. The dataset is designed such that waterbirds appear against a water background and landbirds appear against a land background 100% of the time in the train and validation split while in the test split 50% of the birds are shown against the opposite background. This design displays a large performance drop-off for models that attend to spurious background correlations when evaluated on the test set. Noteworthy is that this approach results in an unusual train/validation/test split of 43%/5%/52%. Especially after removing all unbiased validation examples this resulted in only 600 validation examples left[1].

## 4.2 Evaluation Metrics

Model evaluation involved both classification and localization metrics. For classification, the F1 score was employed. To measure the localization performance, the commonly used Intersection over Union (IoU) score and the Energy-based Pointing Game (EPG) (Wang et al., 2020) score were used.

The IoU score requires a binary mask, while attention is continuous between 0 and 1. Consequently, a threshold has to be determined. The default threshold value of 0.5, specified in the authors code[2], resulted in alarmingly low IoU scores. Therefore, several other strategies were explored to determine a more suitable threshold value. Our resulting adaptive IoU threshold sets the top 10% highest attribution values to one and the rest to zero (see Appendix C for results).

The EPG score computes the fraction of positive attributions within a bounding box (Wang et al., 2020). This metric takes the relative importance of a model's attributions into account as it does not require a binary mask. To investigate whether the model is guided to attend to the object inside a bounding box instead of its background, two novel metrics were utilized. The percentage of attribution inside the segmentation mask is compared to the total attribution in the images (EPG Segmentation) or compared to the total attribution in the bounding box (EPG On-Object). In Appendix F and G a mathematical and visual comparison of these metrics can be found.

### 4.2.1 Best model selection

Since models are jointly optimized for both the classification and localization loss simultaneously, selecting a single best model depends on its use case. A fine-tuned model does not necessarily achieve its best classification and localization performance at the same training epoch. Therefore, for each model configuration (Table 1) an evaluation is performed at every training epoch across the different metrics (F1, IoU, EPG) on the validation set. Given the measured performance at every training epoch, a subset of these checkpoints will be selected that constitute a Pareto-front consisting of only Pareto-dominant models. As shown in Table 1, training is performed with 3 different values for the $\lambda_{loc}$. The resulting Pareto-dominant checkpoints of models trained with different $\lambda_{loc}$ values are combined and evaluated all together on the test set, resulting in one final Pareto-front for every fine-tuned model configuration. In this work, each Pareto-front is calculated in a two-metric way such that the F1 score is evaluated against the EPG or IoU score individually.

## 5 Experiments & Hyper parameters

To support their claims, the authors conducted an extensive overview study. In Table 1 all different model configuration options have been outlined. However, due to time constraints, we will outline per claim the configurations we have tested and with which hyperparameters. All models are optimized using an Adam optimizer (Kingma & Ba, 2014), and a batch size of 64 is employed for all models unless mentioned otherwise.

---

[1]The authors also noticed this distribution but decided to keep it this way to prevent information leakage (Sagawa et al., 2019).

[2]After all experiments had run the authors replied that the threshold was defined per model by maximizing the IoU on a held-out validation set.

Table 1: The possible model configuration, all options (columns) are multiplicative, except for the $\lambda_{loc}$ values which are tied to the corresponding loss functions. This results in 720 possible model configurations. However, only the bold configurations have been tested in this paper.

| Backbone | Attribution method | Guidance depth | Loss | Values of $\lambda_{loc}$ | | |
|---|---|---|---|---|---|---|
| **ResNet 50** | **IxG** | **Input** | **Energy** | $5 \times 10^{-4}$ | $1 \times 10^{-3}$ | $5 \times 10^{-3}$ |
| DenseNet-121 | **B-cos** | Mid1 | **L1** | $1 \times 10^{-3}$ | $5 \times 10^{-3}$ | $1 \times 10^{-2}$ |
| ViT-S | IntGrad | **Mid2** | **PPCE** | $1 \times 10^{-4}$ | $5 \times 10^{-4}$ | $1 \times 10^{-3}$ |
| | GradCam | Mid3 | **RRR*** | $5 \times 10^{-6}$ | $1 \times 10^{-5}$ | $5 \times 10^{-5}$ |
| | | **Final** | | | | |

**Claim 1** states that bounding boxes are sufficient in guiding models. To verify this a ResNet-50 model, pre-trained on ImageNet, is fined-tuned on VOC for 300 epochs using different learning rates ($10^{-3}$, $10^{-4}$, $10^{-5}$). The resulting unguided baseline model with the best F1 score is selected for further fine-tuning with guidance for another 50 epochs with a learning rate of $10^{-4}$. Table 1 displays all evaluated fine-tuned model configurations.

As per **claim 2**, the Energy loss constitutes a good loss function for model guidance with bounding boxes. We test this by comparing its performance to the other loss functions that are tested in claim 1. Furthermore, a qualitative experiment is performed by visualizing the attribution maps of the different loss functions. Here it is investigated how well different loss functions attend to object-level features inside the bounding box. Additionally, we perform a quantitative analysis that assesses how well the model is guided towards an object's segmentation mask relative to its bounding box (see Appendix G). We find this is fundamental to the nature of the paper but only limited attention is given to this in the supplementary material by the authors. For this experiment, we introduced a new pre-processing step that enables the extraction of segmentation masks from images in the VOC dataset alongside their bounding boxes. The specific model configurations trained for this experiment are presented in Appendix B. To evaluate the results, two new evaluation metrics were introduced which are explained in Section 4.2.

In **claim 3** the authors find that model guidance can be accomplished cost-effectively. We investigate this claim by fine-tuning the B-cos baseline model in a similar manner as described for claim 1. However, now training is performed on a dataset where only a fraction (10% or 1%) of the data is annotated with bounding boxes that guide the model, reducing annotation costs. Here we train with a batch size of 16 due to VRAM limitations. Additionally, the robustness of the model to annotation errors is explored since noisy annotation masks are much cheaper to obtain. This is done by fine-tuning the B-cos baseline model using bounding boxes that are dilated by 10%, 25% or 50%. Lastly, an experiment is performed that examines the training efficiency of guided models. Attribution maps extracted from earlier layers are more detailed and as such cost more to compute than attribution maps from later layers. This experiment assesses how training time and model performance are affected when model guidance is performed at various depths.

For **claim 4** the waterbirds dataset is used to verify the ability of guided models to disregard spurious correlations. Three B-cos models were trained: one baseline and two fine-tuned models utilizing the Energy and L1 loss on the input layer, with a $\lambda_{loc}$ of 0.05. After communication with the authors, we found that they fine-tuned their models directly on a ResNet-50 model (pre-trained on ImageNet) instead of on a baseline model when working with waterbirds. All models were trained with a learning rate of $10^{-5}$ using a batch size of 32 instead of 64 owing to VRAM limitations. Furthermore, the training data was augmented by randomly resizing the original images and performing horizontal flips with a 50% chance. Lastly, following the original paper, only 1% of the augmented bounding boxes were used. For each fine-tuned model, the checkpoint that achieved the best accuracy and the one with the best EPG score were selected for evaluation.

### 5.1 Additional Experiment

An additional experiment is performed that aims to assess the class-specific fairness of models utilizing guidance. While the original work shows improved localization performance for guided models, it is unclear

if all categories benefit equally from model guidance. For this we utilize models that are fine-tuned with the configurations presented in Appendix B.

## 5.2 Computational requirements

All models are trained using a single NVIDIA A100 40GB GPU. Table 7 in Appendix D indicates the required model training time. However, all models have been trained multiple times with different hyper-parameters, and the number of epochs differs between all configurations as outlined above. Training all models amounted to approximately 90 hours, and emissions are estimated to be 8.3 kgCO$_2$eq, which is equivalent to driving around 33 km[3]. When comparing training time per epoch we note that fine-tuning with model guidance, especially at earlier layers, requires more training time.

## 6 Results

**Baseline Model Results:** For both the vanilla ResNet-50 model and the B-cos ResNet-50 model the checkpoint with the best validation F1 score was obtained when using a learning rate of $10^{-4}$. Therefore, all fine-tuning with guidance was performed on the baseline models trained with this learning rate. Here we slightly deviate from the authors as, after communication, they reported to have used a learning rate of $10^{-5}$ instead for the vanilla ResNet-50 baseline model.

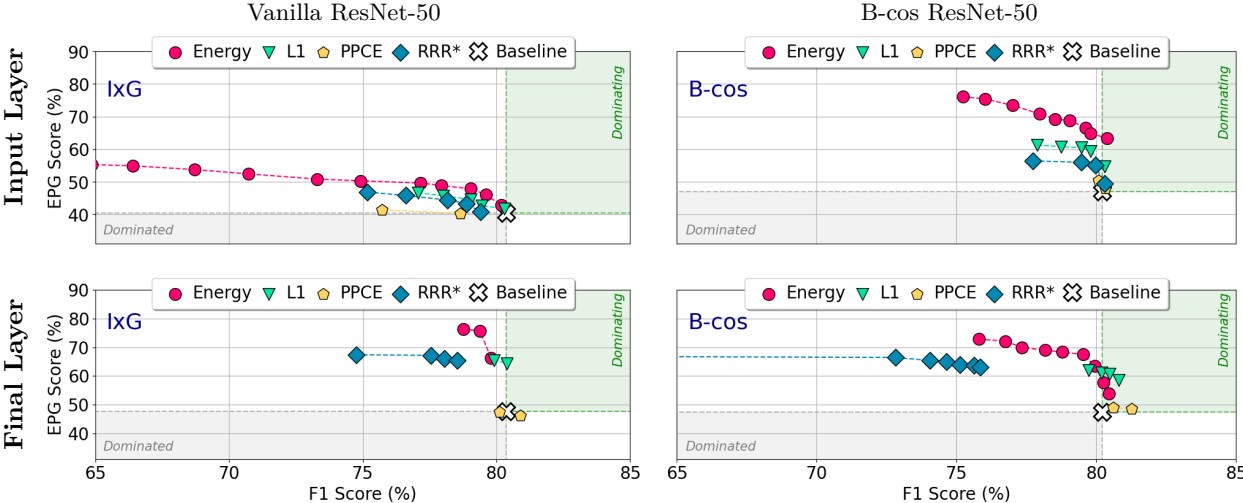

Figure 1: **EPG vs. F1 Pareto-fronts** generated by evaluating model configurations across epochs and $\lambda_{loc}$ for different attribution methods (columns), loss functions (markers), and guiding layers (rows). All models have been fine-tuned from the unguided baseline (white cross). The green (grey) region shows models that strictly dominate (are dominated by) the baseline model.

**Claim 1:** Noteworthy in Figure 1 is that all tested model configurations see an increase in their localization performance when they are fine-tuned using bounding box guidance. Furthermore, the Energy loss achieves the best trade-off between EPG and F1 score as for all configurations its corresponding Pareto-curve dominates the ones obtained from models trained using other loss functions. However, when evaluated based on the IoU score, the L1 loss is most prominent (Figure 2). While these results are generally comparable with those presented by the authors, they show that RRR* achieves localization performance at baseline level when evaluated with the IoU score. Instead, we observe a decrease in localization performance.

**Claim 2:** The results found for claim 1 illustrate that the Energy loss constitutes a good loss function for guiding model attention. Figure 3 further illustrates that models utilizing the Energy loss attend best on

---

[3]Estimations were conducted using the ML Impact calculator presented in Lacoste et al. (2019), using a carbon efficiency of 0.367 kgCO$_2$eq/kWh.

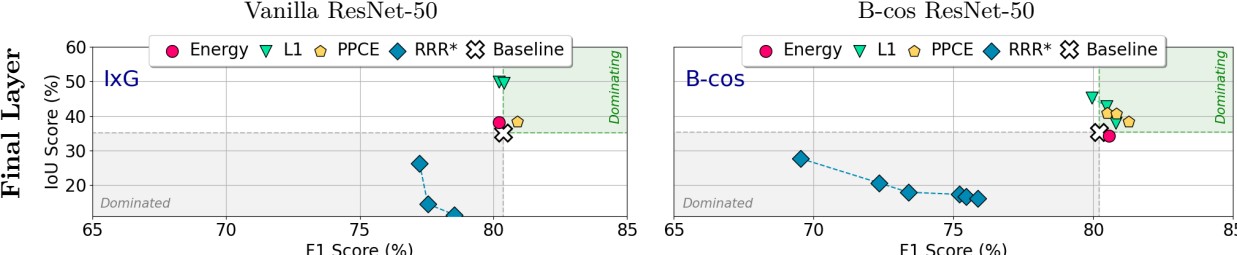

Figure 2: **IoU vs. F1 Pareto-fronts** generated across epochs and $\lambda_{loc}$ for different attribution methods (columns), loss functions (markers), and guiding layers (rows). All models have been fine-tuned from the unguided baseline (white cross). The green (grey) region shows models that strictly dominate (are dominated by) the baseline.

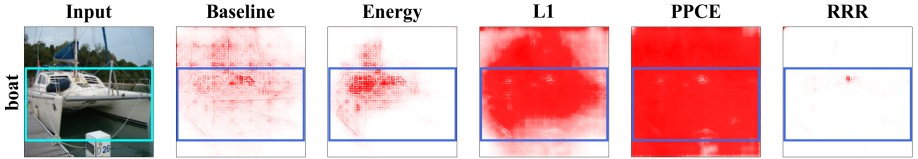

Figure 3: Comparison of input attribution maps for different loss functions (col. 3-6), compared to the unguided baseline input attributions (col. 2). The results are obtained from the last epoch of a finetuned B-cos model utilizing the maximum $\lambda_{loc}$ value. More examples can be found in Appendix H

object-level features. Here it can be observed that the L1 loss, as by design, uniformly spreads attention over the whole bounding box. Lastly, we find that models utilizing the PPCE or RRR* loss perform poorly when compared to the results found by the authors.

Table 2 provides a quantitative perspective that helps separate the performance of the Energy and L1 loss. The Energy loss outperforms the baseline and L1 loss on all metrics. Especially when evaluating the latest training checkpoint we see a significant increase in EPG scores for the Energy loss, while the L1 loss exhibits a much smaller increase. When looking at on-object scores we see that the Energy loss puts more attention inside the segmentation mask especially when compared to the total attention (EPG Segmentation). Lastly, the trade-off between localization and classification performance can be observed. Checkpoints that obtained the best F1 score only increased their EPG score by a few percentage points over the baseline model.

Table 2: **EPG performance** measured on the baseline, and fine-tuned (FT) Energy / L1 loss models. The best F1 model (Best F1) and best localization model (Last) are shown. The **EPG Bounding box** measures the attribution inside the bounding box relative to the image, and **EPG Segmentation** the attribution inside the segmentation mask relative to the image. The **EPG On-Object:** measures attribution inside of the segmentation mask relative to the bounding box. See Appendix G for visual a representation.

|  | Baseline | FT Energy Loss | | FT L1 Loss | |
|---|---|---|---|---|---|
|  | Best F1 | Best F1 | Last | Best F1 | Last |
| **F1 score** | 78.5 | **79.1** | 76.2 | 78.0 | 76.1 |
| **EPG Bounding Box** | 43.2 | 47.9 | **64.8** | 46.6 | 57.2 |
| **EPG Segmentation** | 26.9 | 30.7 | **45.9** | 29.6 | 36.7 |
| **EPG On-Object** | 58.1 | 59.5 | **65.2** | 59.0 | 59.1 |

**Claim 3:** Figure 4 depicts the results for models fine-tuned with only a subset of the training data containing bounding box annotations. Similarly to the authors, we find that models trained with limited annotations still display a significant increase in localization performance over the baseline. Furthermore, in comparison to models that utilized a fully annotated dataset, only a minor performance loss is incurred. Figure 6 shows

that the Energy loss yields consistent results whether the model was fine-tuned with well-made or dilated bounding boxes. However, for the L1 loss a clear decline in localization performance is observed as the dilation of the used bounding boxes is increased. The qualitative results displayed in Figure 7 reinforce these findings. Even with dilated bounding boxes, models fine-tuned with the Energy loss demonstrate improved object focus, whereas models fine-tuned with the L1 loss do not.

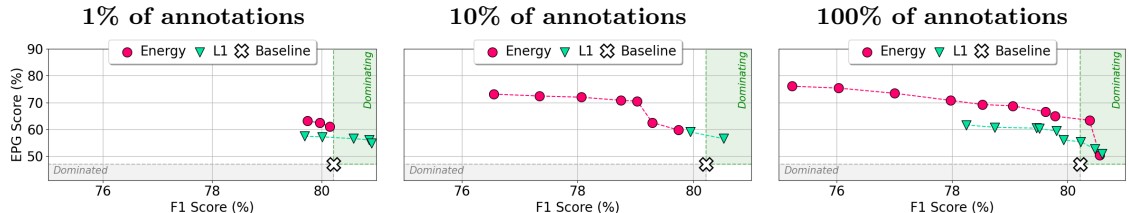

Figure 4: **EPG vs. F1 Pareto-fronts** for a B-cos attribution model fine-tuned from the baseline (white cross), on the input layer using 2 different loss functions (markers). From left to right models are trained using increasingly more bounding box annotations (1%, 10%, and 100%).

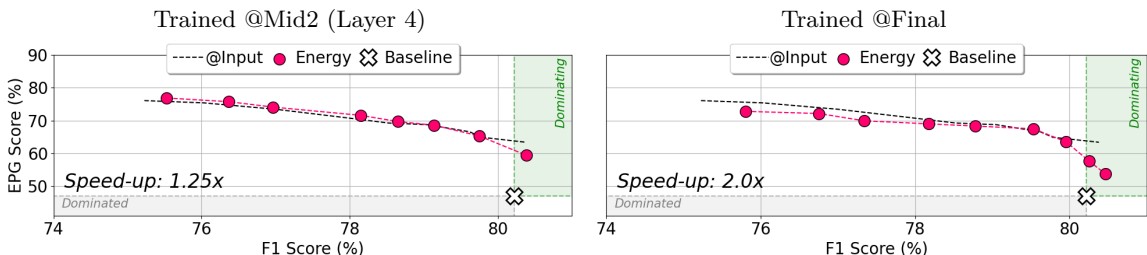

Figure 5: **EPG vs. F1 Pareto-fronts** for a B-cos attribution model fine-tuned from the baseline (white cross) using the Energy loss. A comparison is made between a model guided on the input layer (dashed line) and models trained using the attribution map from the mid-2 (left) and final layer (right). The training-time speed-up over the input layer model is displayed in the bottom left.

**Claim 4:** Table 3 displays the outcomes of the waterbirds experiment. Our findings are in line with the authors and show that guiding can improve model performance on an out-of-distribution test set, with both the L1 and Energy loss surpassing baseline performance. The authors identify classifying waterbirds displayed on a land background as the most difficult examples. However, after inspecting their supplement, it appears that examples of landbirds with a water background yielded even worse results. Therefore, we included both "worst cases". Since the paper doesn't specify which checkpoint is used for evaluation, we report results from the checkpoint with the best localization (last) and classification (best) performance. While we observe a similar trend as in the original paper, our raw accuracy scores differ, with the last checkpoint giving the most comparable results. Additional qualitative results can be found in Appendix J.

Table 3: Waterbirds accuracy scores for baseline and fine-tuned models on different test sets (LoW: Landbird on Water, WoL: Waterbird on land). Results obtained with best accuracy scoring and best localization scoring (last) models.

| Test Dataset | Best Accuracy | | | Last | | |
| --- | --- | --- | --- | --- | --- | --- |
| | LoW | WoL (worst) | Overall | LoW | WoL (worst) | Overall |
| **Baseline** | 27.6 | 45.8 | 65.2 | 38.2 | 40.3 | 68.7 |
| **Energy** | **33.3** | 27.8 | **65.4** | **51.1** | **43.7** | **74.1** |
| **L1** | 26.4 | **52.0** | 65.3 | 36.1 | 37.6 | 67.6 |

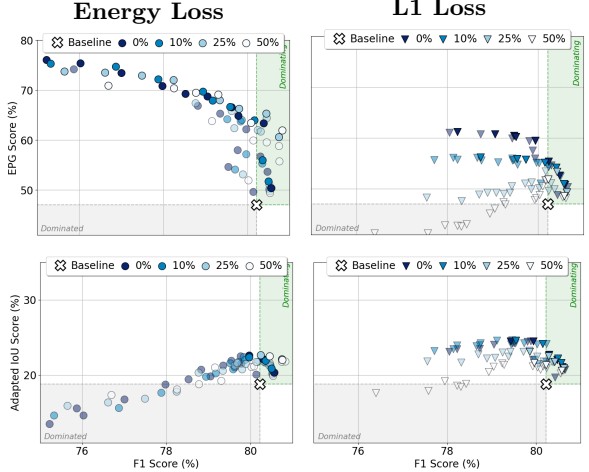

Figure 6: All checkpoints for fine-tuned B-cos models guided at the input layer, using either the Energy or L1 loss and trained with different levels of dilated bounding boxes. The white cross indicates the baseline performance and for each dilation percentage the corresponding Pareto-dominant models are displayed as solid colours.

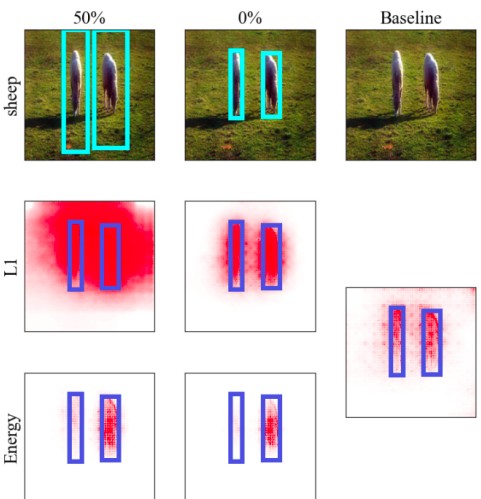

Figure 7: A qualitative comparison between the baseline B-cos model, and B-cos models guided using the Energy or L1 loss at the input layer, trained with or without dilated bounding boxes. More examples can be found in Appendix I.

**Additional experiment:** The fairness results for different classes are illustrated in Table 4. While no correlation can be observed between the number of training samples and the amount of gained localization performance, significant variation can be seen in the performance gained across different classes. For instance, the TV monitor and horse classes both have 144 samples, but compared to their baseline EPG scores differ in gained localization performance by 22.5 percent points. All results are made available in Appendix A.

Table 4: A fairness assessment of the per-class localization performance gains due to model guidance. **Score:** obtained EPG score on the test set. **Gains:** percentage increase in localization performance of fine-tuned models over the baseline. FT best corresponds to the best performing fine-tuned Bcos model based on F1 score, while FT last indicates best model by its localization performance.

| Image Class | Training Samples | Base Model Score | FT Last Score | FT Last Gain | FT Best F1 Score | FT Best F1 Gain |
|---|---|---|---|---|---|---|
| **Aeroplane** | 133 | 57.7 | 65.8 | 14.1 | 61.4 | 6.4 |
| **Bicycle** | 122 | 55.4 | 69.0 | 24.4 | 58.6 | 5.7 |
| **Horse** | 144 | 65.4 | 81.8 | 25.1 | 70.1 | 7.1 |
| **Tvmonitor** | 144 | 27.8 | 41.0 | 47.6 | 29.7 | 6.9 |

## 7 Discussion

Since we obtained similar experimental results to those of the authors in our reproducibility study we can conclude that their work is reproducible. Furthermore, we have found support for all claims that are made. Our results lead us to conclude that bounding boxes are indeed sufficient to achieve model guidance and that the Energy loss constitutes a good loss function for this task. Moreover, while bounding boxes are already cheaper to obtain than segmentation masks, our experiments with dilated and limited bounding boxes demonstrate that model guidance can be accomplished even more cost-effectively. Lastly, as indicated

by the results with the Waterbirds dataset, models trained with limited annotations also display improved model generalization under distributional shifts at test time.

While we find the majority of the authors' work to be reproducible, our results have revealed a few differences and observations. One notable initial difference concerns the vanilla ResNet-50 baseline model utilized in all subsequent fine-tuning experiments. Before communication with the authors, our results indicated that the optimal baseline was obtained by training with a lower learning rate than the authors later communicated to us. Although this seems rather impactful, when evaluating both baseline models insignificant performance differences are observed. This explains why similar results are obtained still.

As described in Section 4.2, due to the lack of information about the threshold value used in the calculation of the IoU metric we deviate from the implementation of the authors. Still, in line with the authors, we find that the L1 loss delivers the biggest increase in performance when its localization performance is evaluated based on the IoU metric. However, we do not find that the Energy loss surpasses the baseline performance and even observe a decrease for the RRR* loss. Therefore, when basing localization performance on the IoU metric many of the claims made don't seem to hold up. However, we attribute most of these inconsistencies to the IoU metric not being a fair and suitable measure to evaluate the continuous model attributions.

The EPG score on the other hand is more suitable for continuous attributions and the results evaluated on this metric are all reproducible. However, much like the results indicate, it seems rather logical that models trained using the Energy loss perform best on the EPG metric as they are directly optimized based on this metric via the corresponding localization loss.

Lastly, in our experiments (Table 2) we found evidence that when models do focus on on-object features this doesn't necessarily lead to improved classification performance. The utilized Pareto-fronts showcase this trade-off between the F1 score and the localization performance, whereas models that perform better on the localization metric do so at the expense of classification performance and vice-versa. This trade-off seems counter-intuitive as one might expect that when a model focuses on relevant features its classification performance would also increase. Furthermore, we observe (Table 4), that when utilizing model guidance, the localization performance does increase unequally across classes, suggesting the models' inability to correctly focus on the relevant features of a diverse set of classes. However, while guidance doesn't necessarily improve classification performance, it does improve model generalization under distributional shifts at test time.

### 7.1 What was easy?

The paper by Rao et al. (2023) is very well written, offering some background and clear explanations for the conducted experiments. Additionally, the supplement provides extra insight and contains the utilized hyperparameters. Furthermore, the codebase worked immediately but the provided environment required a couple of tweaks.

### 7.2 What was difficult?

Although we applaud the authors for their thorough evaluation, given our limited computational resources, we were unable to validate all results. Consequently, we were unable to test on all datasets and with all model backbones. While most of the code was provided, certain parts did not function as expected, preventing us from reproducing the results, as presented in the paper, straight away. For example, when creating Pareto-fronts using the function provided by the authors, a front based on multiple metrics is produced, whereas the plots in the paper depict fronts on a one-on-one metric basis. Therefore adjustments had to be made. The lack of comments in the codebase complicated this process, thus where possible these have been added. Moreover, to further improve reproducibility, the authors could have provided more details about certain parameters. Providing the exact learning rate with which the best-evaluated baseline models were trained, and including which $\lambda_{loc}$ value and model epoch are used for the shown visualisations would be of great help. Lastly, we ran into a strange bug with the provided code that forced us to use a smaller batch size of 16/32 for the experiments run with limited annotations. However, as we obtained similar results to those of the authors, this appeared to have minimal impact.

### 7.3 Communication with original authors

When reaching out to the authors we got a detailed response, albeit late with respect to the timespan of this research. Due to time constraints, not all their answers could be incorporated into our experiments and results. Where possible notes have been added with comments of the authors.

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

## A  Full results of Fairness assessment

Table 5: Improvement gains (relative percentage) in EPG score from the baseline model to the best F1 model (FT Best F1) and best EPG model (last)

| Image Class | Training Samples | Base Model Score | FT Last Score | Gain | FT Best F1 Score | Gain |
|---|---|---|---|---|---|---|
| Aeroplane | 133 | 57.7 | 65.8 | 14.1 | 61.4 | 6.4 |
| Bicycle | 122 | 55.4 | 69.0 | 24.4 | 58.6 | 5.7 |
| Bird | 182 | 42.8 | 63.5 | 48.4 | 49.0 | 14.5 |
| Boat | 87 | 42.2 | 55.1 | 30.6 | 44.7 | 6.0 |
| Bottle | 153 | 16.9 | 22.4 | 32.6 | 18.0 | 6.7 |
| Bus | 100 | 59.9 | 73.0 | 22.0 | 63.5 | 6.1 |
| Car | 402 | 48.8 | 61.0 | 24.9 | 52.3 | 7.0 |
| Cat | 166 | 65.9 | 78.3 | 18.9 | 68.9 | 4.7 |
| Chair | 282 | 25.6 | 38.8 | 51.5 | 28.1 | 9.6 |
| Cow | 71 | 54.6 | 68.3 | 25.2 | 58.6 | 7.5 |
| Diningtable | 130 | 47.9 | 73.3 | 52.9 | 52.6 | 9.7 |
| Dog | 210 | 59.4 | 76.8 | 29.2 | 64.3 | 8.3 |
| Horse | 144 | 65.4 | 81.8 | 25.1 | 70.1 | 7.1 |
| Motorbike | 123 | 62.5 | 73.2 | 17.1 | 66.1 | 5.7 |
| Person | 1070 | 45.6 | 61.5 | 34.9 | 49.6 | 8.7 |
| Pottedplant | 153 | 30.8 | 39.9 | 29.5 | 32.3 | 4.6 |
| Sheep | 49 | 49.1 | 64.2 | 30.8 | 53.4 | 8.8 |
| Sofa | 188 | 52.0 | 67.8 | 30.3 | 56.0 | 7.7 |
| Train | 128 | 62.9 | 77.2 | 22.6 | 67.1 | 6.6 |
| Tvmonitor | 144 | 27.8 | 41.0 | 47.6 | 29.7 | 6.9 |

## B  Hyper parameters Additional Experiment 1/2

Table 6: Hyperparameters used for the additional experiments.

| | Value |
|---|---|
| Dataset | VOC |
| Attribution method | B-cos |
| Loss | Energy / L1 (only Add 1) |
| Backbone | Resnet50 |
| Learning rate | 0.0001 |
| Lambda (FN) | 0.001 |
| Layer | Input |
| Batch-size | 64 |
| Epochs | 300 (Baseline), 50 (Fine-tune) |

# C IoU Thresholding

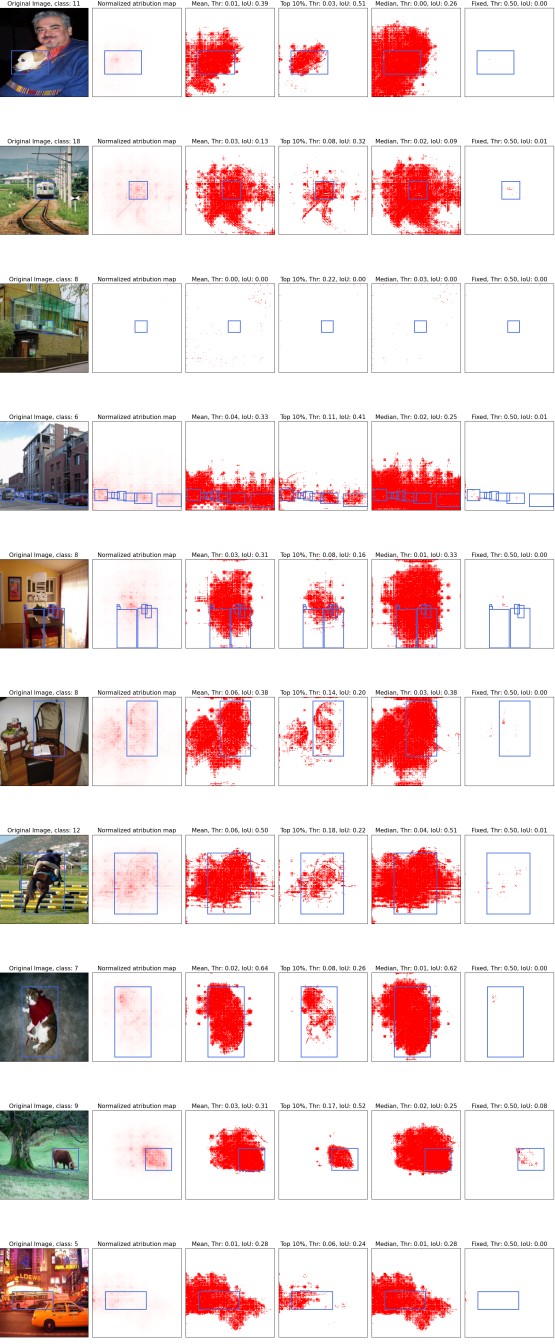

Different methods of setting a threshold (columns 3-6) to create a binary mask from the normalized attribution mask (column 2) extracted at the input layer

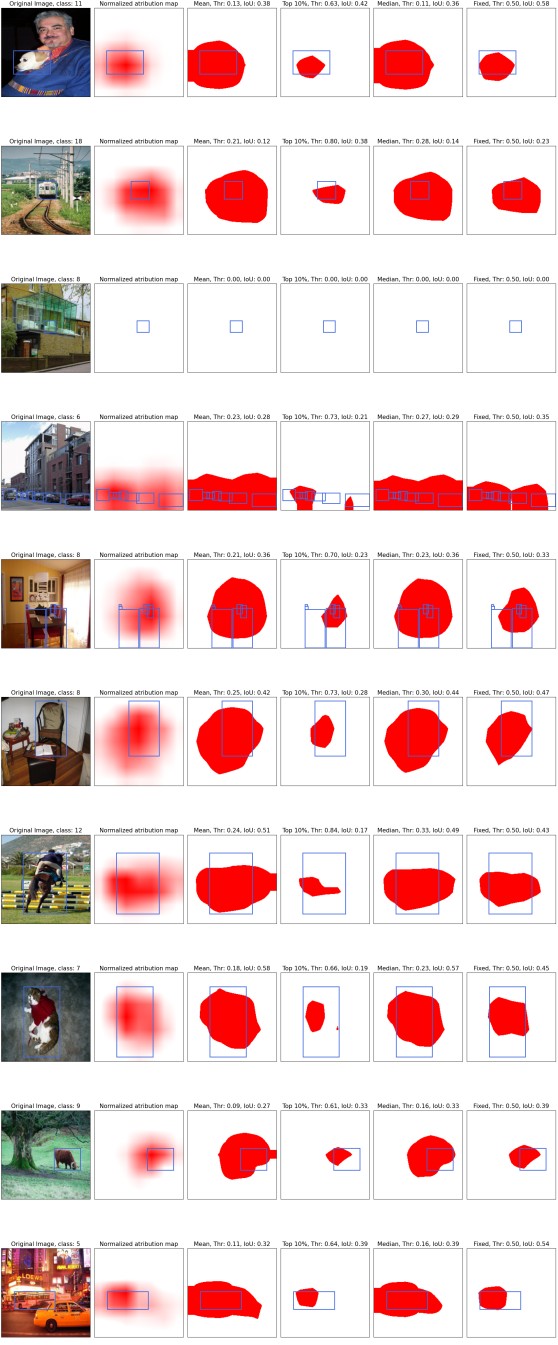

Different methods of setting a threshold (columns 3-6) to create a binary mask from the normalized attribution mask (column 2) extracted at the final layer

## D   Model training times

Table 7: Model training time in hours:minutes:seconds per epoch and in total for different model configurations. Note: Training with dilated annotations did not change training time significantly.

|  | B-cos | | IxG | | Waterbirds | |
| --- | --- | --- | --- | --- | --- | --- |
|  | **Total** | **Epoch** | **Total** | **Epoch** | **Total** | **Epoch** |
| **Baseline** | 2:35:00 | 0:00:31 | 1:05:00 | 0:00:13 | 2:36:00 | 0:00:27 |
| **Finetune** | | | | | | |
| **Input** | | | | | | |
| 100% of annotations (batch size 64) | 1:04:00 | 0:01:17 | 0:45:00 | 0:00:54 | 4:10:00 | 0:00:43 |
| 10% of annotations (batch size 16) | 1:13:00 | 0:01:28 | - | - | - | - |
| 1% of annotations (batch size 16) | 1:04:00 | 0:01:17 | - | - | - | - |
| **Mid2** | 0:51:00 | 0:01:01 | - | - | - | - |
| **Last** | 0:32:00 | 0:00:38 | 0:17:00 | 0:00:20 | - | - |

## E   Loss functions

Outlined below are the loss functions utilized in our experiments

$$W, H = \text{Image size} \qquad\qquad A^+ = \text{Positive attribution map}$$
$$k = \text{Class in image} \qquad\qquad M = \text{Binary bounding box mask}$$

$$\text{L1}_k = \frac{1}{\text{H} \times \text{W}} \sum_{h=1}^{H} \sum_{w=1}^{W} ||M_{k,hw} - \hat{A}^+_{k,hw}||_1 \quad (1) \qquad -\text{EPG}_k = -\frac{\sum_{h=1}^{H} \sum_{w=1}^{W} M_{k,hw} A^+_{k,hw}}{\sum_{h=1}^{H} \sum_{w=1}^{W} A^+_{k,hw}} \quad (2)$$

$$\text{PPCE}_k = -\frac{1}{||M_k||_1} \sum_{h=1}^{H} \sum_{w=1}^{W} M_{k,hw} \log(\hat{A}^+_{k,hw}) \quad (3) \qquad \text{RRR*}_k = \sum_{h=1}^{H} \sum_{w=1}^{W} (1 - M_{k,hw}) \hat{A}^2_{k,hw} \quad (4)$$

## F   EPG metrics

Comparison of the mathematical implementation of the three utilized EPG metrics.

$$W, H = \text{Image size} \qquad\qquad A^+ = \text{Positive attribution map}$$
$$S = \text{Binary segmentation mask} \qquad\qquad M = \text{Binary bounding box mask}$$

$$\text{EPG} = \frac{\sum_{h=1}^{H} \sum_{w=1}^{W} M_{hw} A^+_{hw}}{\sum_{h=1}^{H} \sum_{w=1}^{W} A^+_{hw}}$$

$$\text{EPG Segmentation} = \frac{\sum_{h=1}^{H} \sum_{w=1}^{W} S_{hw} A^+_{hw}}{\sum_{h=1}^{H} \sum_{w=1}^{W} A^+_{hw}}$$

$$\text{EPG On-object} = \frac{\sum_{h=1}^{H} \sum_{w=1}^{W} S_{hw} A^+_{hw}}{\sum_{h=1}^{H} \sum_{w=1}^{W} M_{hw} A^+_{hw}}$$

## G   Visual EPG metrics comparison

Due to the limited availability of segmented data in the VOC dataset, our goal was to maximize the number of samples for evaluation. The original authors conducted a train-validation split on the training set from the VOC dataset and performed testing on the validation set. Therefore, we utilized both the validation and test samples enabling us to gather a diverse set of examples without including samples on which the model has been trained.

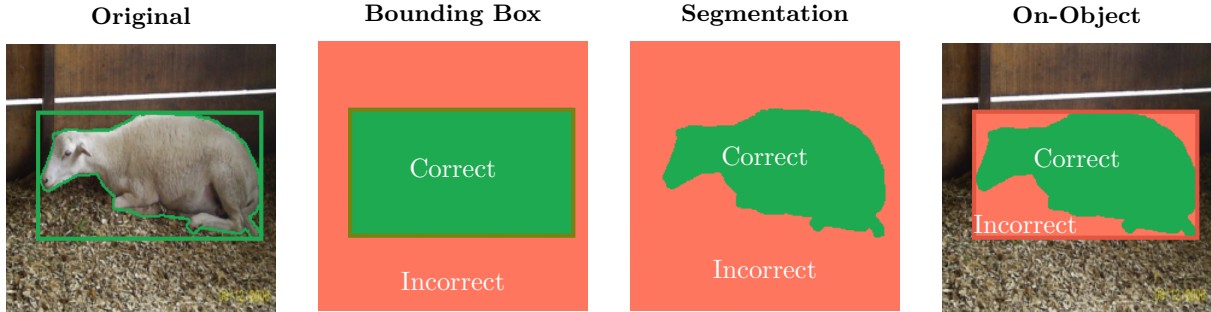

Figure 10: Visual representation of the EPG scores evaluated for Bounding Box, Segmentation and On-object score. Counted as correct is presented in green, Counted as incorrect is presented in red.

## H    Qualatative loss comparisons

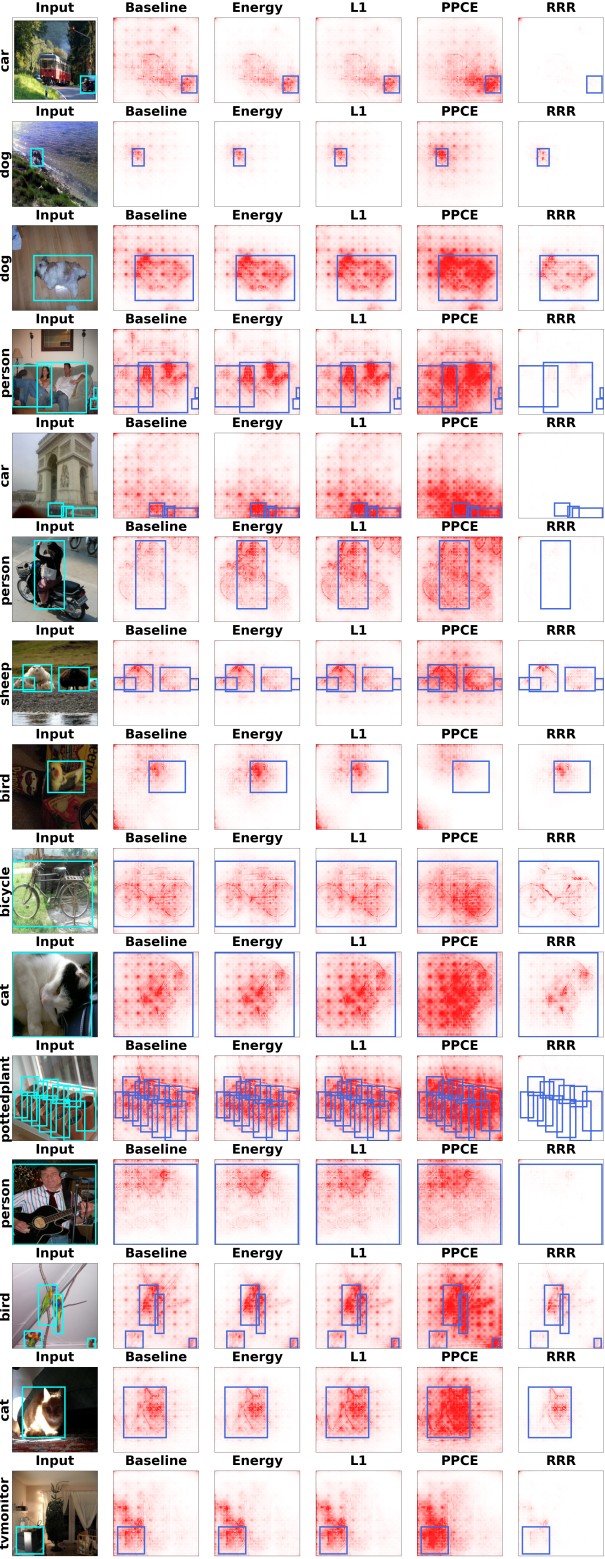

Loss comparisons on input layer

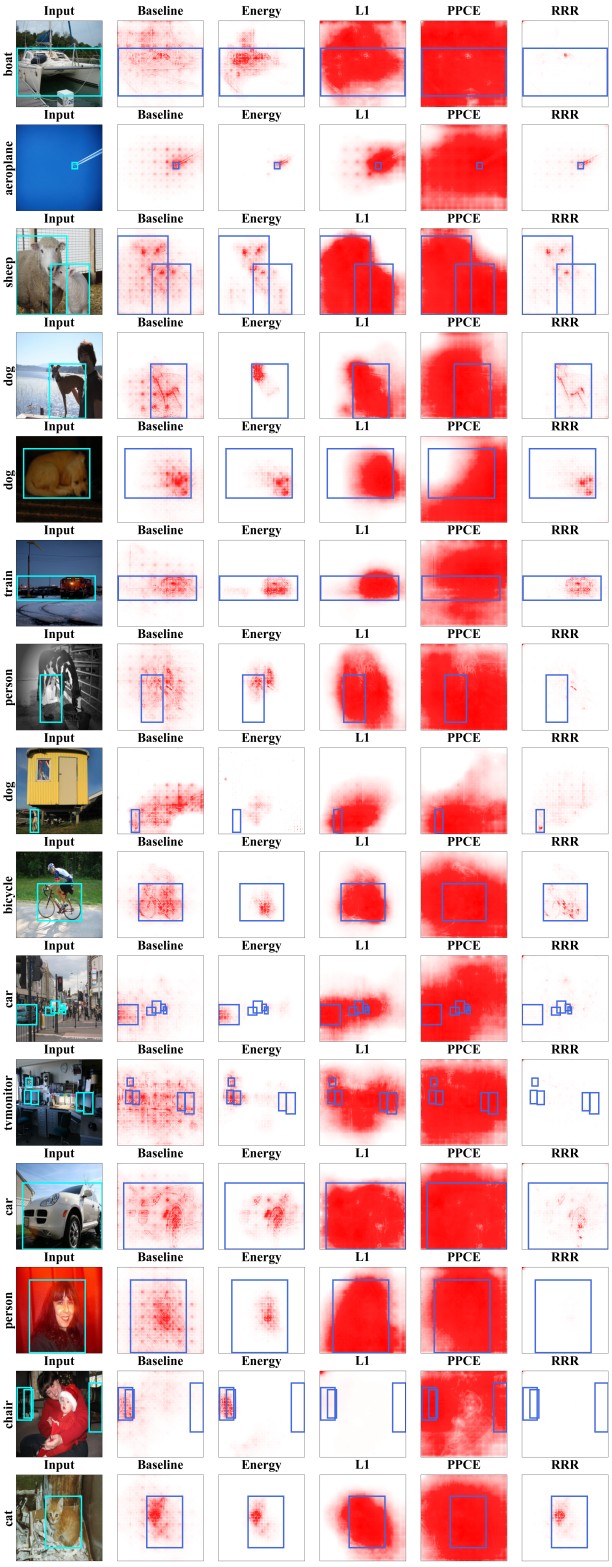

Loss comparisons on final layer

Figure 12: Comparison of attributions at the input and final layers for four different loss functions compared to the unguided baseline.

# I Qualatative dilated bounding boxes

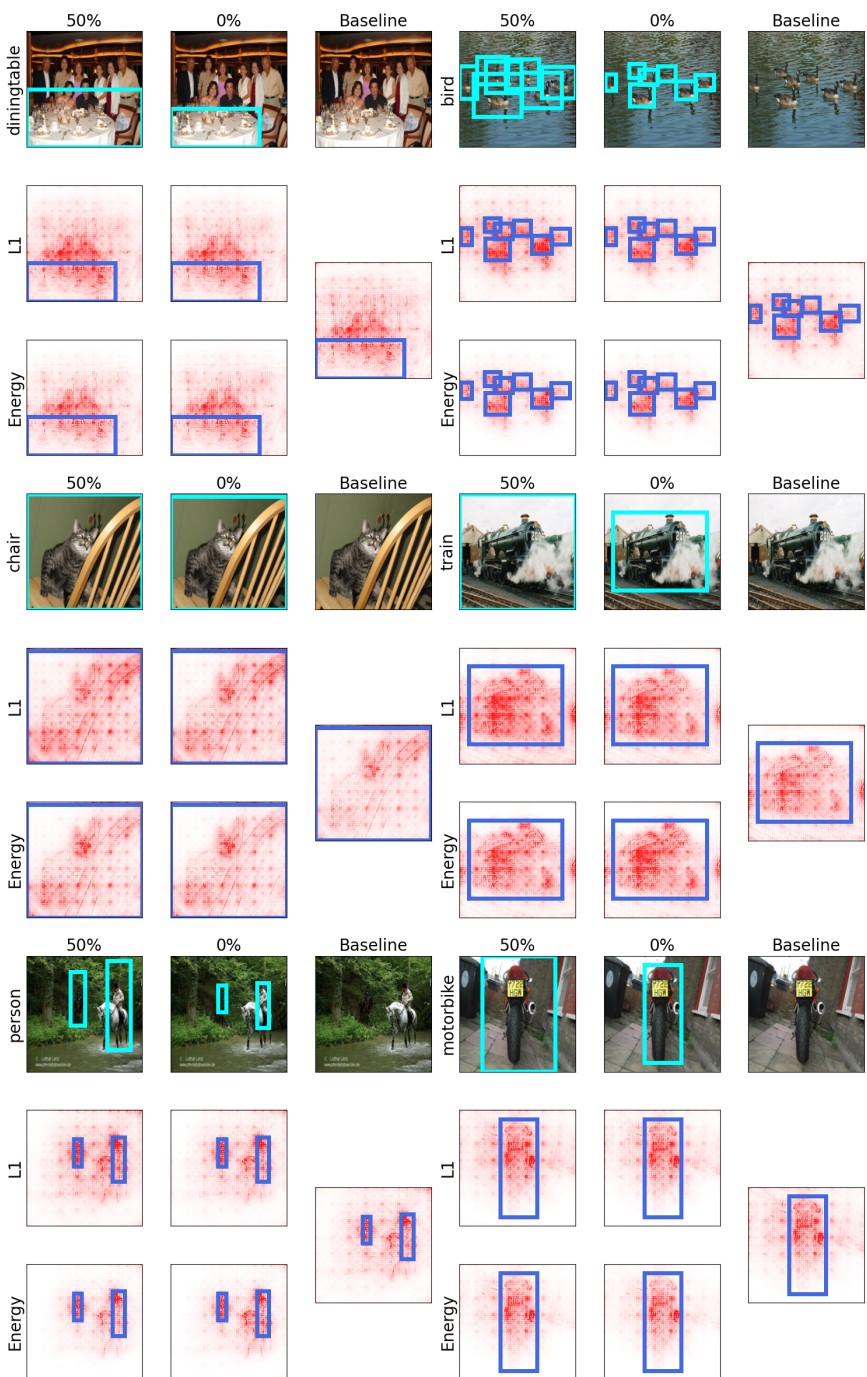

Dilated bounding boxes on input layer

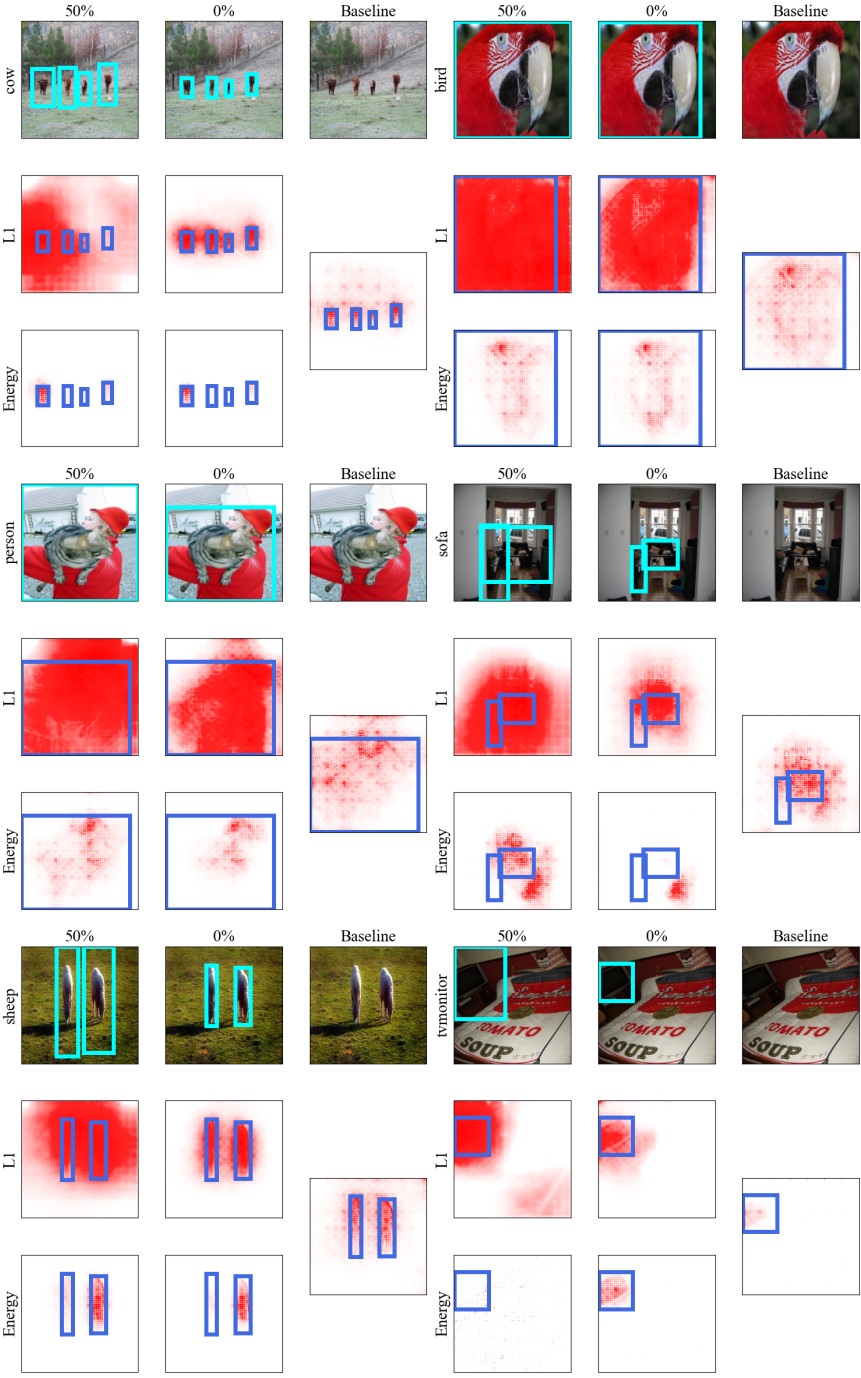

Dilated bounding boxes on final layer

Figure 14: Dilated bounding boxes on attributions made from the input and final layer compared to the baseline

## J   Qualatative waterbird attributions

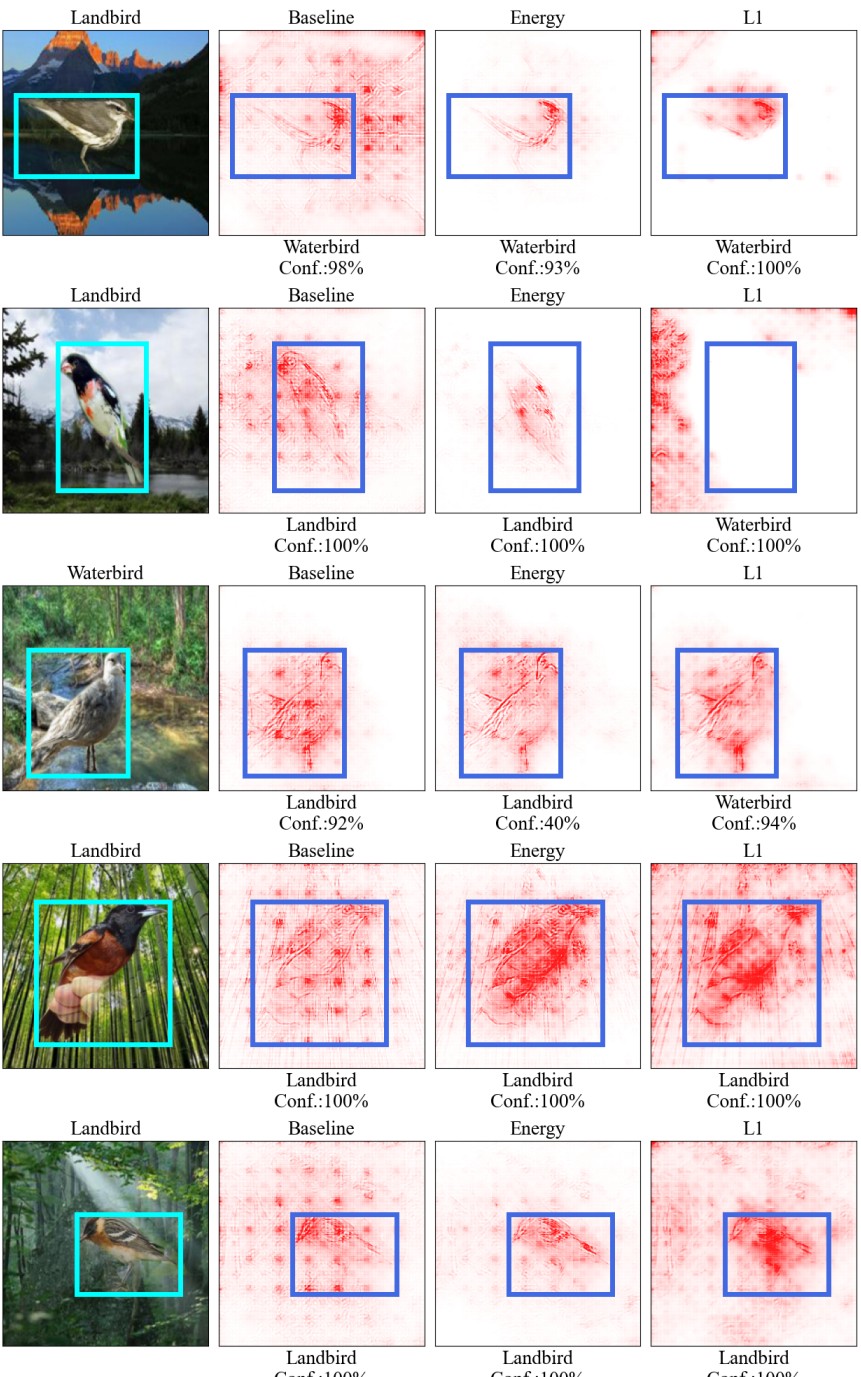

Waterbird attributions on input layer

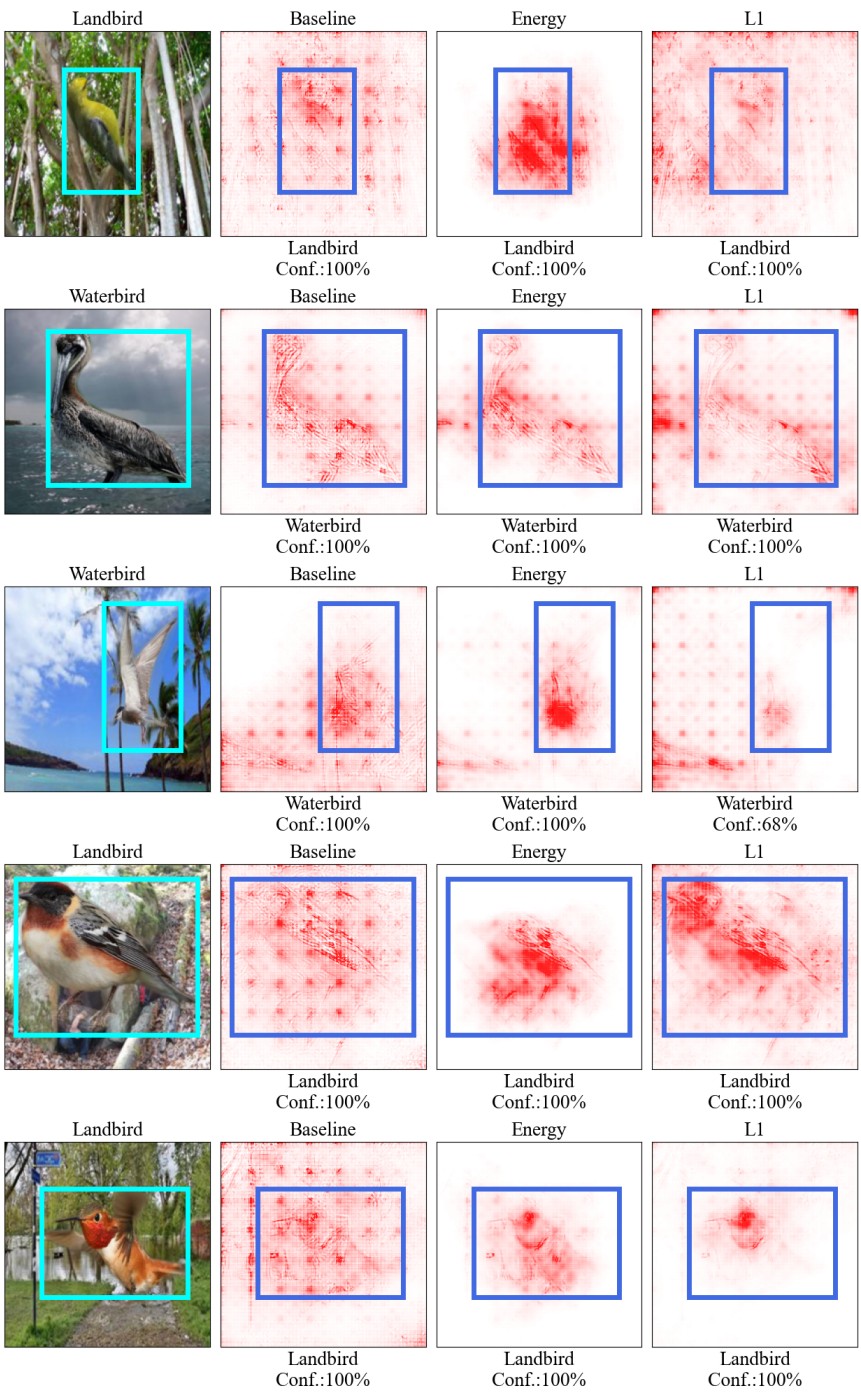

Waterbird attributions on final layer

Figure 16: Waterbird attributions made from the input and final layer

