# OpenReview forum: "Reproducibility Study of “Studying How to Efficiently and Effectively Guide Models with Explanations”"
_TMLR — Rejected by TMLR_

### Review · Reviewer_kPfL · 2024-03-09

**Summary Of Contributions:**

The manuscript presents a reproducibility study of Rao et al. (2023). The manuscript highlights parts from Rao at al. that are not properly described in the paper nor captured in the original codebase.

**Audience:**

Yes

**Broader Impact Concerns:**

No concerns

**Claims And Evidence:**

Yes

**Requested Changes:**

The reviewer is used to evaluate reproducibility papers.

Overall, the paper looks good and seems to be well executed. The reviewer appreciates the effort put by the authors to communicate with the authors of the original study.

One point that was a bit unclear to the reviewer is what is the criteria that the authors use to conclude that the claim from the original paper have been reproduced. For example, when discussing the results, which magnitude of a difference is considered to ensure the reproducibility? Did the authors run any statistical significance between the originally reported numbers and the reproduced numbers?

For the results, it might be worth plotting the results as a difference between original results and the reproduced results.

**Strengths And Weaknesses:**

Strengths:
- The paper is well written and easy to follow.
- The execution looks solid.
- The reviewer appreciates What was easy and what was hard subsections.

Weaknesses:
- It is a bit unclear what is the criteria used by the authors to conclude that the study is reproducible.

---

### Review · Reviewer_wwcc · 2024-03-10

**Summary Of Contributions:**

This paper is a reproducibility study of the experiments presented in Rao et al. 2023. that explores how to guide recognition models efficient and effectively using bounding boxes.
The study shows that apart from some lack of early responses from the original paper's authors, it was able to reproduce most of the experiments in the paper.

**Audience:**

Yes

**Broader Impact Concerns:**

No concerns

**Claims And Evidence:**

Yes

**Requested Changes:**

\- Reproduce results on MS COCO

**Strengths And Weaknesses:**

\+ the study is well written and the presentation of the results is clear.
\+ the additinoal experiments on per-calss fairness added a better understanding of the problem
\+ Even if the paper is overall reproducible (at least on the tested datasets) I liked the insights provided in the discussoin.

\- due to lack of resources, the evalaution was performed only on VOC07 and waterbirds. It would be important to assess the results on a larger dataset such as MS COCO. Obtaiining the desired results on small datasets is always easier than on larger ones.

---

### Review · Reviewer_F6GK · 2024-03-21

**Summary Of Contributions:**

The authors study the reproducibility of "Studying How to Efficiently and Effectively Guide Models with Explanations" (Rao et al.). The autors replicate the experiments of the original paper using published source code, extended some extended code to cover all the original experiments. The authors show that they can replicate the experiments and conclusions of Rao et al.
The authors highlight a few novelties compared to Rao et al.:
* They highlight a difference in the baseline hyperparameters, but show that this does not significantly impact the results.
* They highlight a few differences in conclusions as to which loss to use when using the IoU metric for localization, but underline that this metric is imperfect to study localization because of the thresholding; however the EPG metric conclusions are in line with Rao et Al.
* They highlight that model with better localization do not necessarily perform better for overall (in-distribution) classification performance, and show that the effect is class dependent on the datasets under study.

**Audience:**

No

**Broader Impact Concerns:**

No broader impact concerns.

**Claims And Evidence:**

Yes

**Requested Changes:**

General:
* Improve clarity and references to work performed by the authors, with respect to work performed by Rao et al (2023).
It is better to always mention (Rao et al 2023) in line, rather than using qualifiers such as "In the original study" (p2), or use "the authors" in the third person, which is always quite ambiguous as to whether the authors refer to themselves or to Rao et al.
I recommend the authors use the first-person more in order to highlight their contributions more clearly:
"However, certain crucial parts were missing from the codebase and have thus been added. " -> We have added...
"Therefore, several other strategies were explored" -> We explored several other strategies...

* Since Rao et al. provide references R1-R9 for their observations, it would make sense to refer to these, rather than introduce a new summarized numbering (Claims 1-4).

Misc:
Citation for the Waterbirds dataset is missing when introducing it on top of page 3. In that same paragraph, footnote 1 is unclear: if the authors want to use the same standard Waterbirds-100 dataset, it is clear that its distribution should not be changed.

**Strengths And Weaknesses:**

The main weekness of this replication study, is that while it replicates the experiments of Rao et Al, it does not clearly identifying major new findings or contradictions with respect to the original work.

The acceptance criteria of TMLR (https://jmlr.org/tmlr/acceptance-criteria.html) gives a hypotetical example of "A [...] report that re-runs the experiments of a published paper has educational value [but is] unlikely to be of interest to (even a subset of) the TMLR audience". This seems to apply to this work, which runs a subset of the experiments of the original work, without bringing new generalizable and convincing conclusions.

The authors recognize that their findings are in line with Rao et Al. They do attempt to bring new insights into the fact that in-distribution classfication performance is not always improved by model guidance. However, this is already outlined in the original work: see (Rao et al., R6) "R6 Model guidance can improve accuracy. [...] However, overall we observe a trade-off between localization and accuracy"

In short, while I laud the authors efforts to validate the experiments and conclusions of Rao et Al, I do not believe that this replication warrants a publication in TMLR on its own in its current state. Indeed, while the claims seem accurate and supported by evidence, they do not seem to bring significant insights over Rao et Al.

---

### Decision · Action_Editor_VbCw · 2024-04-30

**Recommendation:** Reject

**Comment:**

Although the reviewers lauded the authors' effort to reproduce the original work, they emphasized that the lack of significant new analysis or insights makes the work insufficient for acceptance at this time. The authors did not take the opportunity to respond to the reviewer questions and comments during the discussion period, suggesting they may generally agree with the sentiment in the reviews. If the authors undertake a major revision that includes substantial additional analysis and insights, the paper could be appropriate for publication at TMLR.

**Audience:**

No. Although the paper completes a reproducibility study of a significant proportion of the original paper, the reviewers are in general agreement that the paper does not provide additional insights to a level that would be of interest to a TMLR audience.

**Claims And Evidence:**

Yes.

**Resubmission Of Major Revision:**

The authors may consider submitting a major revision at a later time.